# Effects of Service Quality, Corporate Image, and Customer Trust on the Corporate Reputation of Airlines

**HakJun Song [1], Wenjia Ruan [1] and Yunmi Park [2],***

[1] Department of Hotel & Convention Management, Pai Chai University, 155-40 Baejae-ro, Seo-gu, Daejeon 302-735, Korea; bloodia@pcu.ac.kr (H.S.); ruanwenjia@pcu.ac.kr (W.R.)

[2] Department of Aviation Service, Cheong Ju University, 298 Daesung-ro, Cheongwon-gu, Cheong Ju 28503, Korea

* Correspondence: ympark@cju.ac.kr; Tel.: +82-43-229-7871

**Abstract:** With the continuously increasing demand for air travel, competition among airlines has increased as well. The purpose of this study is to examine empirically the causal relationships among the perceived service quality, corporate image, customer trust, and corporate reputation of Asiana Airline in South Korea using SERVQUAL measures. An onsite survey of Korean passengers was conducted. The results of this study reveal that responsiveness and reliability of service quality significantly affect corporate image and customer trust, whereas tangibles, empathy, and assurance of service quality are not significant antecedents of corporate image and customer trust. It is also found that corporate image and customer trust significantly affect corporate reputation. Based on these findings, theoretical importance and practical implications are discussed in this paper.

**Keywords:** perceived quality; corporate image; customer trust; corporate reputation

---

## 1. Introduction

The airline industry is one of the dominant service sectors in the world, serving 4.3 billion passengers who flew on 46.1 million flights in 2018, and at the beginning of 2019, revenue passenger kilometers (RPKs) rose 5.3% compared with that of 2018 [1,2]. With the continuously increasing demand for air travel, competition among airlines has increased as well, because passengers have become sophisticated shoppers for alternative options that offer better value for their money [3]. Among the various marketing variables, service quality was proven to be one of the key factors in increasing the competitive advantage of airlines [4,5]. Providing high-quality service is an important marketing requirement, as it creates opportunities for airlines to build a positive corporate image and earn trust from passengers, thus retaining customer patronage [6,7].

One purpose of providing better quality is to meet customers' expectations and retain customer patronage. In prior studies, it was generally confirmed that good service quality can lead to customer satisfaction and loyalty [8,9]. Another main purpose of providing better quality is to improve the competitiveness of an airline. A better corporate image was also proven to be an outcome of high service quality, and a favorable corporate image also helped customers to understand products better and decrease his/her uncertainty during the decision-making process [10]. Reputation was considered to be the most valuable intangible asset for a company to maintain a sustainable competitive advantage [11,12]. A positive corporate reputation can lead to desirable consequences, such as increased cash flow and profitability [11,12]. However, the formation of corporate reputation is a long-term process, involving the accumulation of stakeholders' judgments about a firm over time [13]. Herbig, Milewicz, and Golden state that corporate reputation is built through a corporation's credible

actions [14]. Furthermore, it has been stated that corporate reputation is affected by corporate image and customer trust [15,16]; however, there has been little empirical research that has integrated service quality and three important corporation aspects—corporate image, customer trust, and corporate reputation—together to explore the process of corporate reputation building. Accordingly, the main purpose of the current research is to determine the influence of airline service quality on corporate reputation by using SERVQUAL and reputation quotient (RQ) measures.

## 2. Literature Review

### 2.1. Perceived Service Quality

Perceived service quality refers to the discrepancy between subjective expectation and the actual perception of the performance of a service by customers, which is used to measure the degree to which a provided service matches customers' expectations [4,17]. Expectation is a customer's belief concerning the provided service, while performance is a customer's subjective view of a provided service level [18,19]. Perceived service quality has been suggested as a main competitive factor to keep a product/company/brand at a competitive advantage in the market [20], as it was revealed to be a critical measurement of business performance and a company's long-term viability [21].

SERVQUAL is a common scale to measure perceived service quality, which consists of five factors, namely tangibles, responsiveness, reliability, empathy, and assurance. SERVQUAL was first put forward by Parasuraman, Zeithaml, and Berry [4] and then applied to measure the perceived service quality of some specific industries, products, and target markets [8,9,21]. It was also introduced into the airline industry with a set of attributes of airline services to measure the perception of service quality of passengers [22]. Alkhatib and Migdadi identified the key operational determinants of airline service quality across regions, flight ranges, and destinations through multi-regression multi-level analysis. The results showed that the strongest impact of the operational determinants on airline service quality was in Arab, the weakest impact was in Europe, and there was no significant impact in Africa [22].

Service quality may affect the image of carriers [23,24]. Chang and Yeh [25] demonstrated that airline service quality was an important determining factor in an airline's competitive advantage, possibly determining whether potential passengers will choose a certain airline or not. The perceived service quality of an airline also affects the perceived value, satisfaction, airline choice, and behavioral intention of passengers [9,26].

Therefore, in order to occupy a place in the competitive airline market, it is particularly significant to provide high-quality service to passengers [27].

### 2.2. Image

Corporate image is defined as customer's subjective perception of a product, brand, or company that offers products and services [28]. The antecedent factors of corporate image have been examined in previous studies. Bullmore [29] demonstrated that advertisements are the main medium for brands or corporations to deliver images to consumers. Hu, Kandampully, and Juwaheer [30] revealed that satisfaction, perceived service quality, and perceived value offered by a company are the important determinants of corporate image. It was also determined that corporate social responsibility could enhance a product, brand, or company's image [31,32]. In turn, corporate image can affect customers' behavioral intentions and the perception of the operation of the company as well [28,33].

Thus, corporate image can be seen as an essential factor in an airline study, because it can affect customers' behavioral intentions, as well as customers' overall evaluation of a company.

### 2.3. Trust

Trust is an important construct in various academic fields, especially in organizational theory and marketing [34,35]. In a given exchange relationship, trust is defined as the level of reliability guaranteed by the seller to the buyer, and it was proven to be an important factor in maintaining

a positive relationship between the buyer and seller in marketing [34,36]. Customer trust can be increased when company performance exceeds a customer's expectation by keeping its promises and building strong exchange relationships between customers and companies [26,37,38]. Because it was revealed that service quality positively affects customer trust [34,39], perceived uncertainties in a service provider's relationship with customers can be reduced by satisfying customers' expectations by providing better service quality. Moreover, trust can maintain customer loyalty [40], and it was proven to be a mediating variable between corporate social responsibility (CSR) and customers' behavioral intention [41].

Therefore, consumer trust toward an airline corporation is also essential, as it can reduce perceived uncertainty and risk and improve customers' perceptions of company performance.

### 2.4. Corporate Reputation

Corporate reputation has always been a concern due to turmoil in the business environment, increasing customers' expectations and pressure from various stakeholder groups calling for a company to manage its reputation well [42]. Fombrun and Shanley [43] assert that corporate reputation is an important intangible resource that may affect corporate performance and even survival. Corporate reputation has been related to brand equity and corporate credibility in the field of marketing [13,44]. In this study, corporate reputation can be considered to be an overall view of the company [45]. Reputation is an essential way for companies to maintain a sustainable competitive advantage and endure a long term relationship with multiple stakeholder groups [11]. It was also demonstrated in prior studies that corporate reputation is affected by customer trust [46]. In turn, a favorable corporate reputation has a positive impact on the decisions of an organization's key stakeholders (such as customers, creditors, and employees) and their attitudes and behaviors towards the company, making them predisposed in favor of the company [43,47].

In terms of measuring corporate reputation, there are two main ways, namely single-overall measurement to make a generic measurement of corporate reputation and multi-face measurement to measure corporate reputation through overall perception. In general, multi-face measurement is usually used, as it can measure the specific perceptions of all stakeholders' various elements. Fombrun, Gardberg, and Sever [48] initially proposed the reputation quotient, which was widely used to measure corporate reputation, and many subsequent measures were also developed on this basis. In reputation quotient, corporate reputation is measured through six dimensions, namely product quality, vision and leadership, financial performance, social and environmental responsibility, workplace environment, and emotional appeal. Reputation quotient was proven to be a good measurement, which can be applicable to different cultural contexts and commercial settings through a general and broad consideration of most stakeholder groups [49,50]. For example, Del-Castillo-Feito, Blanco-González, and González-Vázquez employed reputation quotient to examine the relationship between image and reputation in the setting of universities. Moreover, reputation quotient was measured through performance, innovation, social responsibility, services, governance and workplace climate, which was modified based on the research object [51].

### 2.5. Hypotheses Development

#### 2.5.1. Relationship between Service Quality and Corporate Image

Corporate image is considered to be an important factor in evaluating a company, which also affects customers' perception of provided services and his/her choice of companies [10]. Grönroos [33] demonstrated that the establishment of corporate image was mainly through two indices—technical quality, which is provided to customers through the service experience, and functional quality, which is the manner of providing the service. Park, Robertson, and Wu [9] revealed that service quality, perceived value, and customer satisfaction can affect corporate reputation. The overall image of a company may be heightened when customers perceive service quality through repeated service

encounters [52]. Yang, Hsieh, Li, and Yang [53] took the case of low-cost carriers (LCCs) in Taiwan and revealed that when perceived service quality increases, an airline's image will improve as well. Although some studies attempt to reveal the casual relationship between perceived service quality and corporate image, few studies have adopted SERVQUAL measures.

Thus, we would like to suggest the hypothesis that each factor of airline service quality (tangibles, responsiveness, assurance, empathy, and reliability) will have a positively significant influence on corporate image.

**Hypothesis 1.** *Perceived service quality has a positively significant effect on corporate image.*

1-1: Tangibles have a positively significant effect on corporate image.
1-2: Responsiveness has a positively significant effect on corporate image.
1-3: Reliability has a positively significant effect on corporate image.
1-4: Empathy has a positively significant effect on corporate image.
1-5: Assurance has a positively significant effect on corporate image.

### 2.5.2. Relationship between Service Quality and Customer Trust

Perceived service quality plays a crucial role in shaping customers' views on the level of service quality. Customer trust is related to the assessment of a provided service, as it was demonstrated that when customers perceive high service quality, their trust of the service provider is deepened as well [54–56]. In other words, when customers have a better perception of service quality, they are more likely to trust the service provider. As perceived service quality is the discrepancy between customer expectations and actual perception of service performance, in order to meet customer expectations of service, corporations will strive to reduce perceived uncertainties, fluctuations, and risk in the relationship between service providers and customers and win customer trust over time [34,57].

Building on the research above, this paper intends to analyze the impact of perceived service quality on customer trust towards airlines. Hence, we hypothesize the following:

**Hypothesis 2.** *Perceived service quality (tangibles, responsiveness, assurance, empathy, and reliability) has a positively significant effect on customer trust.*

2-1: Tangibles have a positively significant effect on customer trust.
2-2: Responsiveness has a positively significant effect on customer trust.
2-3: Reliability has a positively significant effect on customer trust.
2-4: Empathy has a positively significant effect on customer trust.
2-5: Assurance has a positively significant effect on customer trust.

### 2.5.3. Relationship between Corporate Image and Corporate Reputation

The casual relationship between corporate image and cooperate reputation remains unclear. Porter [58] suggests that a good reputation is the antecedent to rebuilding an innovative image in the industry, whereas Allen [15] proposes that corporate reputation is the result of establishing a corporate image. However, Fombrun [10] explains that corporate reputation starts from corporate identity, which is the internal stakeholder's perception of the organization. Additionally, corporate reputation is formed through all the images developed by different stakeholders. Allen [15] demonstrated that corporate reputation is the outcome of the overall corporate image building process. Moreover, building a favorable image is also revealed to be important, because it can develop a good reputation and enhance customer trust [59].

With regard to airlines, when passengers perceive that the corporate image is favorable, his/her perception of corporate reputation is enhanced as well. Therefore, we hypothesized that corporate image positively and significantly affects corporate reputation as follows:

**Hypothesis 3.** *Corporate image has a positively significant effect on corporate reputation (vision and leadership, finance performance, corporate social responsibility, and workplace environment).*

3-1: Corporate image has a positively significant effect on vision and leadership.
3-2: Corporate image has a positively significant effect on finance performance.
3-3: Corporate image has a positively significant effect on corporate social responsibility.
3-4: Corporate image has a positively significant effect on workplace environment.

2.5.4. Relationship between Customer Trust and Corporate Reputation

Customer trust is viewed as customers' believability regarding a company, which is determined by whether a company really does what it promises to do. In other words, trust is gained with the congruence between the message and action of a company. Trust is an essential factor in the success of a corporation, as it was revealed that corporate reputation is built through its credible actions. Delivering promised quality is important to building a positive reputation. Because credibility is fragile, there is a high cost to regain credibility once it is lost [13]. Herbig and Milewicz [13] demonstrated that a company can build a favorable reputation by fulfilling its promise, whereas failing to meet its expressed intention may damage corporate reputation. Customer satisfaction and product quality was proven to affect corporate reputation in prior studies [60]. Trust was also found to be the antecedent factor of corporate reputation [16,46].

Based on previous studies, customer trust can be seen as the antecedent of corporate reputation (vision and leadership, finance performance, corporate social responsibility, and workplace environment). Thus, we hypothesize the following (Figure 1):

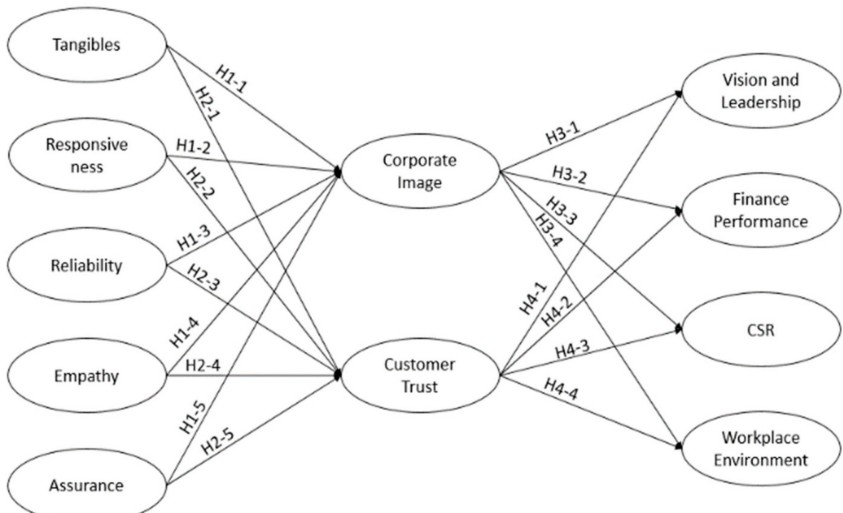

**Figure 1.** Hypothetical Model. CSR: corporate social responsibility.

**Hypothesis 4.** *Customer trust has a* positively significant *effect on corporate reputation.*

4-1: Customer trust has a positively significant effect on vision and leadership.
4-2: Customer trust has a positively significant effect on finance performance.
4-3: Customer trust has a positively significant effect on corporate social responsibility.
4-4: Customer trust has a positively significant effect on workplace environment.

## 3. Methodology

### 3.1. Measurement

A preliminary list of measurement items was selected after an extensive review of the literature pertaining to the service quality of airlines [2,4,61], corporate image [41,62], customer trust [41,46], and corporate reputation [42,48].

To evaluate customers' perceptions and ensure the validity of the questionnaire, tourism scholars and airline managers were invited to comment on the selected items. A pilot test was conducted with airline managers, flight attendants, and graduate students during January 2018. In the pilot test, 50 valid questionnaires were collected, which helped to refine measurement items for clarity. All measurement items used a five-point Likert scale (1 = strongly disagree and 5 = strongly agree).

### 3.2. Data Collection and Analysis

With regard to a specific data collection process, an onsite survey was conducted from March to May 2018 in three international airport lounges, namely Incheon International Airport, Gimpo International Airport, and Gimhae International Airport of Korea, targeting Korean passengers who have flown with Asiana Airlines. Asiana Airlines was selected as the research object as it is one of two representative airline companies in Korea (i.e., Korean Airlines and Asiana Airlines), with high popularity and good quality service. Seven field researchers briefly explained the purpose of this study and invited contacted passengers to participate in the survey. The subjects received a self-administered questionnaire after the researchers obtained their consents. The respondents received a small gift as a token of appreciation for their participation after returning the questionnaire. Of the 385 questionnaires distributed, 355 questionnaires were collected, representing a response rate of 92.2%. A thorough examination eliminated 27 questionnaires due to insincere responses and missing data. The final analysis had a dataset of 328 questionnaires. Structural equation modeling (SEM) was run using an R statistic program. A two-stage testing procedure was adopted in this study [63]. In the first stage, confirmatory factor analysis (CFA) was used to estimate the measurement model for the research variables. In the second stage, the structural model was used to test hypothetical relationships among the latent variables [63].

## 4. Result

### 4.1. Demographic Characteristics

As shown in Table 1, the proportion of female respondents (51.2%) was slightly higher than that of male respondents (48.8%). Most respondents had attended institutions of higher education, including university (48.8%), 2-year college (17.7%), and graduate school (17.4%). The dominant age category was 40–49 years (50.6%) followed by the age category of 30–39 years (32.0%). Most respondents were married (79.3%) compared with those who were unmarries (19.5%). The majority of married people had children (75.9%). A monthly income range between KRW 3 and 4.9 million (US $1 is about equivalent to KRW 1138) was dominant (34.8%), followed by KRW 2–2.9 million (18.9%). Expert or technician and office staff were the predominant categories of professions at 29.0% and 27.7%, respectively, followed by service provider (15.5%) and housewife (10.1%).

### 4.2. Measurement Model

To confirm whether the data in this study violated the assumption of multivariate normality, Mardia's standardized coefficient was employed. These data showed multivariate normal distributions, because Mardia's standardized coefficient for the measurement model (59.671) in this study was greater than the criterion of 5 [64]. Because the data did not meet the assumption, the robust maximum likelihood method was used to estimate the structural equation modeling [64,65].

**Table 1.** Demographic characteristics of respondents (N = 328).

| Characteristic | N (%) | Characteristic | N (%) |
|---|---|---|---|
| *Gender* | | *Marital status* | |
| Male | 160 (48.8) | Single | 64 (19.5) |
| Female | 168 (51.2) | Married | 260 (79.3) |
| | | Other | 4 (1.2) |
| *Education level* | | *Age* | |
| Primary school | 0(0) | Under 20 | 6 (1.8) |
| Junior/Senior school | 53 (16.2) | 20–29 | 26 (7.9) |
| 2-year college | 58 (17.7) | 30–39 | 105 (32.0) |
| University | 160 (48.8) | 40–49 | 166 (50.6) |
| Graduate school | 57 (17.4) | 50–59 | 22 (6.7) |
| | | Over 60 | 3 (0.9) |
| *Monthly income level* | | *Occupation* | |
| Less than 1 million won | 21 (6.4) | Expert or technician | 95 (29.0) |
| 1–1.9 million won | 57 (17.4) | Businessman | 22 (6.7) |
| 2–2.9 million won | 62 (18.9) | Service | 51 (15.5) |
| 3–4.9 million won | 114 (34.8) | Office staff | 91 (27.7) |
| 5–5.9 million won | 41 (12.5) | Civil servant | 6 (1.8) |
| More than 6 million won | 33 (10.1) | Housewife | 33 (10.1) |
| *Children* | | Student | 5 (1.5) |
| No | 79 (24.1) | Retired | 4 (1.2) |
| Yes | 249 (75.9) | Others | 21 (6.4) |

The measurement model was found to fit the data well, with the following goodness-of-fit indices from CFA: normed fit index (NFI) = 0.873, non-normed fit index (NNFI) = 0.907, comparative fit index (CFI) = 0.923, and root-mean-square error of approximation (RMSEA) = 0.063 (see Table 2). Table 2 also presents Cronbach's alpha values that were used to estimate the reliability of the multi-item scales: tangibles (TAN; 0.844), reliability (REL; 0.856), responsiveness (RES; 0.896), assurance (ASS; 0.896), empathy (EMP; 0.923), image (IMG; 0.916), trust (TRU; 0.902), vision and leadership (VL; 0.905), workplace environment (WE; 0.875), corporate social responsibility (CSR; 0.887), and finance performance (FP; 0.895). All alpha coefficients were above the conventional cut-off point of 0.7 [66], which indicates an acceptable level of reliability for each construct.

Additionally, to confirm the convergent and discriminant validity of the constructs, further statistics were analyzed. To ensure the convergent validity of the constructs, the average variable extended (AVE) and composite reliability values of the multi-item scales must be greater than the minimum criterion of 0.5 and 0.7, respectively [67]. As shown in Table 3, all the AVE and composite reliability values indicated that the measurement model has adequate convergent validity. Three methods were used in the study to check the discriminant validity of the constructs.

The first method holds that AVE should exceed the corresponding squared correlation of variables. The results in Table 3 show that the highest squared correlation between some variables was greater than the AVE for the corresponding inter-constructs [69]. Therefore, the other two methods, namely confidence intervals and constrained models, could be used to confirm the discriminant validity. The former calculated the confidence interval of correlation by adding or subtracting two standard errors of correlation between the constructs. The discriminant validity was supported if the confidence interval did not include the criterion of 1.0 [70]. For example, the squared correlation between REL and TAN (0.669) was greater than the AVE for TAN (0.646) and REL (0.667). However, the discriminant validity was supported by using the confidence interval of correlation among the latent variables, because the confidence interval of correlation between REL and TAN was (0.758, 0.878), which did not include the criterion of 1.0. It is noteworthy that the discriminant validity between WE and VL was not confirmed, because the squared correlation between WE and VL (0.825) was greater than the AVE for the corresponding inter-constructs (WE = 0.700, VL = 0.766) and the confidence interval of

correlation between WE and VL (0.812, 1.004) included the criterion of 1.0. However, it was finally found that the discriminant validity between WE and VL could be supported with the method of the constrained model as the S-B Chi-square difference test statistic between the constrained model and the un-constrained model exceeded the criterion of 3.84 with one degree of freedom (chi$^2$(1) = 202.112, $p < 0.001$) [71,72].

**Table 2.** Reliability and confirmatory factor analysis.

| Constructs | Factor Loading | *t*-Value | Cronbach's Alpha |
|---|---|---|---|
| **Tangibles (TAN)** | | | **0.844** |
| Asiana Airlines aircraft appearance and in-flight facilities are visually appealing. | 0.795 | 28.796 | |
| The staff of Asiana Airlines are clean and tidy. | 0.817 | 33.800 | |
| Asiana Airlines' equipment is very modern. | 0.799 | 33.955 | |
| **Reliability (REL)** | | | 0.856 |
| Asiana Airlines keeps its promise to customers accurately. | 0.793 | 32.171 | |
| Asiana Airlines will provide service that is exactly what it should provide. | 0.838 | 38.962 | |
| Asiana Airlines strives to provide error-free services. | 0.819 | 34.293 | |
| **Responsiveness (RES)** | | | 0.896 |
| Asiana Airlines reservations' ticketing is fast. | 0.843 | 36.891 | |
| The handling of baggage loss and damage by Asiana Airlines is fast. | 0.903 | 59.111 | |
| Asiana Airlines reservations can be changed and canceled quickly. | 0.841 | 26.096 | |
| **Assurance (ASS)** | | | 0.896 |
| Asiana Airlines staff assures customers. | 0.827 | 35.370 | |
| Asiana Airlines staff is courteous. | 0.878 | 54.352 | |
| Asiana Airlines staff has sufficient working knowledge. | 0.886 | 53.680 | |
| **Empathy (EMP)** | | | 0.923 |
| Asiana Airlines staff shows careful attention to customers. | 0.880 | 45.040 | |
| Asiana Airlines employees strive to meet customer service expectations. | 0.896 | 61.215 | |
| Asiana Airlines staff understand and handle customers' specific requirements. | 0.909 | 62.979 | |
| **Image (IMG)** | | | 0.916 |
| Asiana Airlines is a friendly and thoughtful company. | 0.877 | 46.896 | |
| Asiana Airlines is a clean and wholesome company. | 0.888 | 45.344 | |
| Asiana Airlines is a reliable company. | 0.894 | 55.417 | |
| **Trust (TRU)** | | | 0.902 |
| Asiana Airlines strives to keep its promise to customers. | 0.882 | 49.992 | |
| Asiana Airlines is operating steadily. | 0.856 | 42.347 | |
| Asiana Airlines is fully meeting my expectations as an air transportation company. | 0.872 | 47.388 | |
| **Vision and Leadership (VL)** | | | 0.905 |
| Asiana Airlines has excellent leadership in the industry. | 0.890 | 47.365 | |
| Asiana Airlines has a clear vision for the future. | 0.907 | 66.681 | |
| Asiana Airlines recognizes and responds well to changes in the air transport industry in advance. | 0.826 | 31.038 | |
| **Workplace Environment (WE)** | | | 0.875 |
| Asiana Airlines is well managed. | 0.835 | 39.442 | |
| Asiana Airlines is a good company to work for. | 0.876 | 49.074 | |
| Asiana Airlines is a company with excellent employees. | 0.796 | 25.720 | |
| **Corporate Social Responsibility (CSR)** | | | 0.887 |
| Asiana Airlines is giving support to good things (e.g., social responsibility activities). | 0.823 | 32.442 | |
| Asiana Airlines is very interested in environmental protection. | 0.865 | 44.142 | |
| Asiana Airlines respects human values. | 0.868 | 39.796 | |
| **Finance Performance (FP)** | | | 0.895 |
| Asiana Airlines has a high investment value. | 0.875 | 42.682 | |
| Asiana Airlines is financially more stable than its competitors. | 0.873 | 43.144 | |
| Asiana Airlines is likely to grow in the future. | 0.840 | 36.041 | |

| Measurement model | $\chi^2$ | *Df* | $\chi^2/df$ | NFI | NNFI | CFI | RMSEA |
|---|---|---|---|---|---|---|---|
| Fit indices | 1017.810 | 440 | 2.313 | 0.873 | 0.907 | 0.923 | 0.063 |
| Suggested value * | | | ≤3 | | ≥ 0.9 | ≥ 0.9 | ≤ 0.08 |

Note: Total Cronbach's Alpha: 0.974. Mardia's standardized coefficient = 59.671. All standardized factor loadings are significant at $p < 0.001$. * Suggested values were based on Hair [67] and Bearden, Sharma, and Teel [68].

*4.3. Hypothesis Testing*

Based on the results of the SEM analysis shown in Figure 2, the proposed structural model was found to fit the observed data well with the following goodness-of-fit indices: $\chi^2 = 1076.737$, df = 461; NFI = 0.865; NNFI = 0.906; CFI = 0.918; and RMSEA = 0.063. The proposed hypotheses were tested by evaluating the path relationships among the constructs. The explained variance in endogenous constructs was 53.7% for corporate image, 68.7% for customer trust, 73.9% for vision and leadership,

59.1% for finance performance, 66.9% for CSR, and 82.6% for workplace environment (see Figure 2). The results of the hypotheses testing indicated that among five variables of service quality, only responsiveness and reliability had a significantly positive effect on corporate image and customer trust ($\beta_{RES \to IMG}$ = 0.255, $t$ = 2.316, $p < 0.05$; $\beta_{REL \to IMG}$ = 0.474, $t$ = 3.456, $p < 0.001$; $\beta_{RES \to TRU}$ = 0.505, $t$ = 5.402, $p < 0.001$; $\beta_{REL \to TRU}$ = 0.324, $t$ = 2.070, $p < 0.05$).

**Table 3.** Reliability and validity for the measurement model.

| | TAN | REL | RES | ASS | EMP | IMG | TRU | VL | WE | CSR | FP |
|---|---|---|---|---|---|---|---|---|---|---|---|
| **TAN** | **0.646** | | | | | | | | | | |
| **REL** | 0.818 (0.669) | **0.667** | | | | | | | | | |
| **RES** | 0.742 (0.551) | 0.811 (0.658) | **0.745** | | | | | | | | |
| **ASS** | 0.757 (0.573) | 0.818 (0.668) | 0.772 (0.596) | **0.747** | | | | | | | |
| **EMP** | 0.700 (0.491) | 0.792 (0.626) | 0.708 (0.501) | 0.903 (0.815) | **0.801** | | | | | | |
| **IMG** | 0.596 (0.356) | 0.696 (0.485) | 0.652 (0.426) | 0.629 (0.395) | 0.590 (0.348) | **0.786** | | | | | |
| **TRU** | 0.693 (0.480) | 0.746 (0.556) | 0.786 (0.618) | 0.658 (0.433) | 0.644 (0.415) | 0.754 (0.569) | **0.757** | | | | |
| **VL** | 0.677 (0.458) | 0.664 (0.441) | 0.617 (0.381) | 0.607 (0.369) | 0.582 (0.339) | 0.835 (0.697) | 0.778 (0.606) | **0.766** | | | |
| **WE** | 0.646 (0.417) | 0.686 (0.471) | 0.701 (0.492) | 0.666 (0.443) | 0.654 (0.428) | 0.870 (0.756) | 0.836 (0.699) | 0.908 * (0.825) | **0.700** | | |
| **CSR** | 0.620 (0.385) | 0.652 (0.426) | 0.611 (0.373) | 0.550 (0.302) | 0.539 (0.291) | 0.797 (0.636) | 0.734 (0.539) | 0.865 (0.748) | 0.810 (0.656) | **0.727** | |
| **FP** | 0.604 (0.365) | 0.655 (0.429) | 0.558 (0.311) | 0.591 (0.349) | 0.607 (0.369) | 0.729 (0.532) | 0.725 (0.525) | 0.850 (0.723) | 0.837 (0.701) | 0.820 (0.672) | **0.745** |
| **CR** | 0.845 | 0.857 | 0.897 | 0.898 | 0.924 | 0.917 | 0.903 | 0.907 | 0.875 | 0.889 | 0.745 |

Note: * Pairs of constructs having highest correlations. Numbers on the diagonal indicate average variable extracted (AVE) of latent constructs. Numbers in the parenthesis indicate squared correlation among latent constructs. TAN = Tangibles; REL = Reliability; RES = Responsiveness; ASS = Assurance; EMP = Empathy; IMG = Image; TRU = Trust; VL = Vision and Leadership; WE = Workplace Environment; CSR = Corporate Social Responsibility; FP = Finance Performance.

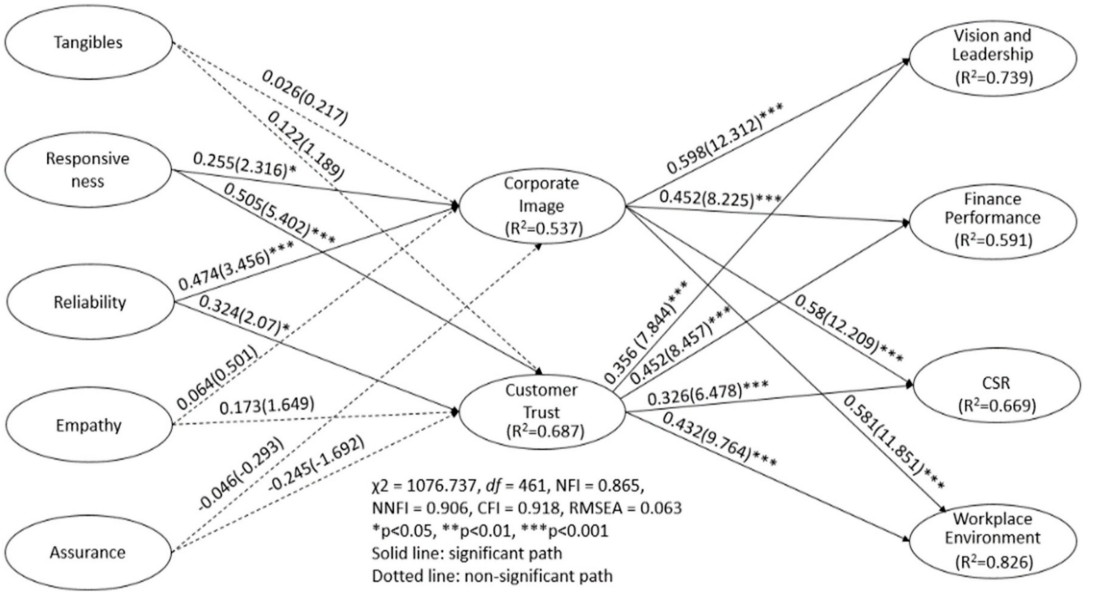

**Figure 2.** Results of the research model.

Moreover, corporate image had a significantly positive impact on all factors of corporate reputation ($\beta_{IMG \to VL} = 0.598$, $t = 12.312$, $p < 0.001$; $\beta_{IMG \to FP} = 0.452$, $t = 8.255$, $p < 0.001$; $\beta_{IMG \to CSR} = 0.580$, $t = 12.209$, $p < 0.001$; $\beta_{IMG \to WE} = 0.581$, $t = 11.851$, $p < 0.001$). Customer trust also had a significantly positive impact on all factors of corporate reputation ($\beta_{TRU \to VL} = 0.356$, $t = 7.844$, $p < 0.001$; $\beta_{TRU \to FP} = 0.452$, $t = 8.457$, $p < 0.001$; $\beta_{TRU \to CSR} = 0.326$, $t = 6.478$, $p < 0.001$; $\beta_{TRU \to WE} = 0.432$, $t = 9.764$, $p < 0.001$).

## 5. Conclusions and Discussion

### 5.1. Conclusions

The influence of airline service quality on passengers' purchasing decision-making processes and the impact of corporate image and customer trust on customers' decision-making behavior have been well explained in previous studies [9,73]. However, little research has been conducted on how perceived service quality affects corporate reputation in the context of airlines. To address this gap, this study employed the scale of SERVQUAL to reveal the process of improving corporate reputation by considering the corporate image and customer trust of Korea's Asiana airline.

The results of the current study indicate that, among the five service quality dimensions, responsiveness and reliability dimensions had a significant effect on corporate image and customer trust, whereas tangibles, empathy, and assurance had an insignificant effect on corporate image and customer trust. In addition, both corporate image and customer trust positively influenced four dimensions of corporate reputation. Thus, the findings of this study confirm that some aspects of airline service quality affect corporate reputation through corporate image and customer trust. It is plausible that the findings will be valuable in subsequent studies for further understanding the building of corporate reputation with respect to airlines.

### 5.2. Discussion

The findings suggest that among the five dimensions of perceived service quality, only responsiveness and reliability had a significant effect on corporate image and customer trust. Notably, the relationship between corporate image and reliability appears to be stronger than that between corporate image and responsiveness, whereas responsiveness puts the greatest impact on customer trust. In the SERVQUAL scale, "reliability" was found to be the most important dimension [61], as well as a well-performed attribute in different airlines, such as Taiwan Airlines and Malaysian Airlines [74]. The result was in line with prior studies that showed that reliability is a more effective and powerful attribute in building corporate image, as well as a main factor that may distinguish full-service carriers (FSCs) and LCCs [2]. Thus, keeping promises to customers and providing error-free service is essential in building a favorable corporate image, especially as it was stated by Asiana Airlines that the company would devote its utmost efforts to fulfilling its commitment to customers by providing unparalleled quality products, services, and technologies, which is in line with the reliability of service quality [75].

Pakdil and Aydın [76] pointed out that "responsiveness" got the highest score in the expectation and perception of all service quality dimensions, which proved the importance of responsiveness in airline service quality. Moreover, findings on the significant effect of responsiveness on customer trust suggest that providing efficient check-in services and the responsiveness of the crew with respect to passengers' requests and complaints contribute to fostering trust for the company among customers. Asiana Airlines adopts trust-based management to provide customers with quick and comfortable service, earning and maintaining the trust and respect of each customer [75]. This goal may aid in improving customer trust for an airline, as quick responsiveness of service can be perceived directly by customers throughout the process, from before the flight to after the flight. Consequently, airline companies should prioritize quick responses to maximize service quality and earn customer trust over time.

Moreover, it was found that several perceived service qualities affect corporate reputation through corporate image and customer trust. This finding implies that customers' perceptions of service

quality directly contribute to generating or improving the airline corporate image and customer trust and indirectly contribute to building airline corporate reputation. Therefore, in practice, it is crucial to establish a favorable corporate image of friendliness, thoughtfulness, cleanliness, and reliability, as well as to gain customer trust by keeping promises and meeting expectations of customers, because good service quality can establish favorable corporate reputation through corporate image and customer trust.

Furthermore, corporate image was verified to be a stronger predictor of each factor of corporate reputation. A favorable corporate image is likely to encourage customers' positive assessments of a company and gain favor [77]. Therefore, airline managers should prioritize and allocate resources for developing better service quality and improving airline crews' service–encounter performance, especially responsiveness and reliability, in order to build a better corporate image, for example keeping promises to customers accurately, providing services without mistakes to ensure the reliability of service, and paying more attention to passengers' requests, including ticket reservations, baggage loss and damage, ticket changes, and cancellations. The eventual success of an airline is proven to be dependent on whether it has the ability to deliver high-quality service consistently to its customers [55].

Airlines aim to establish and maintain a favorable corporate reputation so as to leverage reputation for competitive advantage. A company that engages in large-scale, corporate social responsibility activities and has high media exposure will build a more favorable corporate reputation than its counterparts [43,78,79]. Asiana Airlines, which is among the top two representative airline companies in Korea, initiated a charity program named Change for Good in 1994, in order to support the United Nations Children's Fund (UNICEF). Passengers were encouraged to donate any loose change voluntary at the end of each flight for this charity. By early 2015, the program had raised over $9.6 million for UNICEF. This activity may help in improving its corporate reputation [41]. A good corporate reputation is an intangible asset of a corporation that may provide valuable returns to the corporation, such as increasing customers' purchase intentions; attracting and retaining excellent employees, as well as reputable suppliers; and even safeguarding the organization in times of crises [42,80]. Based on these findings, corporations should focus on customer orientation and provide services with better quality so as to enhance their corporate reputation [60,81]. In other words, to build a favorable corporate reputation with an enjoyable working environment, stable finance performance, desirable vision and leadership, and a high level of social responsibility and to eventually improve the quality of life for all stakeholders, a corporation should build a desirable image and strive to earn and maintain trust by keeping its promises and providing unparalleled quality services to every customer.

*5.3. Limitations and Future Research Trends*

Although the results of this study contribute to the corporate reputation literature in general and specifically for the airline industry, there are still some limitations of this study that require future research. First, because the object of this study is Asiana Airline—one of the top two airlines in South Korea—the results may not be applicable to the entire airline industry in Korea. Second, only four dimensions were used in the reputation quotient, emotional appearance was not considered when measuring corporate reputation. In future studies, a comparison between FSCs and LCCs should be conducted to obtain a comprehensive understanding of the airline corporation reputation building process.

**Author Contributions:** Investigation, Y.P.; Methodology, H.S.; Supervision, H.S.; Writing – original draft, W.R.; Writing – review & editing, H.S.

**Conflicts of Interest:** The authors declare no conflict of interest.

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
