# Peer review of "Effects of Service Quality, Corporate Image, and Customer Trust on the Corporate Reputation of Airlines"

_sustainability, doi:10.3390/su11123302_

Round 1
Reviewer 1 Report
Thanks the Journal for the opportunity to review this paper and thanks the authors for your work.
The manuscript is interesting but I found some points that need attention.
Major considerations
Title: The authors should consider to reduce the title. “Asiana Airline of South Korea” is one element that could be eliminated from the title considering the expected reproducibility that any scientific research should have. Please, also consider eliminating SERVQUAL from the title because it is only the way of measuring service quality and it does not represent any innovation in this research. In fact, there are hundreds of researches that used before the SERVQUAL scale.
Introduction: The purposes expressed in lines 40-45 at the end of introduction are not reached. This is a weak point that must be seriously reword for clarity. First, the theoretical model is not validated because some hypotheses will be rejected. Second, is partially reached because service quality is a multidimensional construct and only some dimensions are confirmed in the empirical study. Third, the article does not analysed mediation effects. For this point the authors could consider following Mallinckrodt et al. (2006) for instance.
Literature Review: I have found some incongruencies in the theoretical framework when it is compared with the posterior section where hypotheses are developed. The first one is related to corporate reputation. The final section devoted to corporate reputation finishes presenting a scale for measuring it with six dimensions and one of them is surprisingly quality of service (Lines 111-123). This is an incongruency because the theoretical model relates the perceived quality of the service with corporate reputation that includes at the same time the quality of the service. But this is not the only problem. Later, Hypothesis 3, that is divided into 4 sub-hypotheses (Lines 184-187), forgets service quality (the first dimension in Lines 111-123), forgets leadership (only remains vision), and finally the hypothesis 3 forgets emotional appeal.
Hypotheses: I have some additional concerns about Hypotheses. Although there are doubts about the universality of the five classical dimensions (Buttle, 1996), service quality has been recognized as a second-order construct (Babakus and Boller, 1992). However, the theoretical model does not considers service quality as a construct and instead considers each of the five dimensions as single constructs. That is completely different that what is suppose to do, the model is not measuring the effect of service quality in the other variables in the model. The model is measuring the effect of each dimension of service quality (according to SERVQUAL scale) to the rest of variables. Thus, title, purposes, theory (Section 2.4) and model development (Section 2.5) are not really congruent.
Please, consider service quality as a second order construct or change the rationale of the manuscript to explain and justify what already is tested with the model. For information about second order constructs see for instance Johnson et al. (2011), Van Riel et al. (2017) and for SERVQUAL as second order construct see Han and Baek (2004).
Results: I do not recommend explaining in detail the three methods to test discriminant validity when only the third one is giving the authors good results. I will offer directly the third method to test discriminant validity. Even maintaining the three methods, it is not logical to offer table 3 containing information that is not supporting discriminant validity. Please, remove table 3 and instead offer a table with information related to method number three.
Hypotheses Testing: Figure 2 has not good quality, please improve it.
Minor changes
Line 29 – Please rewrite for clarity. May be full stop after “patronage” should be enough.
Lines 34-35 – This strong assertion needs at least one reference.
Lines 38-40 – In my opinion citation should be at the end of the sentence as follows: Herbing et al. stated that corporate reputation was built through its credible actions [14].
Line 50 – Rewrite for clarity. May be including “the” in the sentence would be good: In which degree “the” provided service…
Line 56 – SERVQUAL is already a known scale, it was and it is. I think that the correct would be to say “is”.
Line 60-61 – Please explain the results obtained in citation number [20] because it seems to be relevant for this study.
Line 218-219 -Please explain the pilot test with details.
Line 231 – Please explain the representativeness of the sample.
Line 234 – Please explain how do you detect insincere responses.
Line 252 – I wonder whether this table is relevant in the paper considering that there are not analyses testing moderation effects of gender, education or income. In addition, in the table I wonder whether language could be corrected to avoid sexism. For instance, businessman could be also businesswoman and, may be, there is an alternative word for housewife.
I hope my comments should help the authors to upgrade their work. All the best.
References used in this review:
Babakus, E., & Boller, G. W. (1992). An empirical assessment of the SERVQUAL scale. Journal of Business research, 24(3), 253-268.
Buttle, F. (1996). SERVQUAL: review, critique, research agenda. European Journal of marketing, 30(1), 8-32.
Han, S. L., & Baek, S. (2004). Antecedents and consequences of service quality in online banking: An application of the SERVQUAL instrument. Advances in Consumer Research, 31, 208-214.
Johnson, R. E., Rosen, C. C., & Chang, C. H. (2011). To aggregate or not to aggregate: Steps for developing and validating higher-order multidimensional constructs. Journal of Business and Psychology, 26(3), 241-248.
Mallinckrodt, B., Abraham, W. T., Wei, M., & Russell, D. W. (2006). Advances in testing the statistical significance of mediation effects. Journal of Counselling Psychology, 53(3), 372.
Van Riel, A. C., Henseler, J., Kemény, I., & Sasovova, Z. (2017). Estimating hierarchical constructs using consistent partial least squares: The case of second-order composites of common factors. Industrial Management & Data Systems, 117(3), 459-477.
Author Response
We sincerely appreciate the reviewers’ comments and suggestions on the previous version of this manuscript. We have thoroughly studied them and have revised the manuscript accordingly. This report summarizes our responses to all the comments (in red for your convenience).
1. The authors should consider to reduce the title. “Asiana Airline of South Korea” is one element that could be eliminated from the title considering the expected reproducibility that any scientific research should have. Please, also consider eliminating SERVQUAL from the title because it is only the way of measuring service quality and it does not represent any innovation in this research. In fact, there are hundreds of researches that used before the SERVQUAL scale.
Response: Thanks for your keen observation. The title has been simplified to focus on the topic of this paper, and related keywords have been deleted as well.
=> “Effects of service quality, corporate image and customer’s trust on corporate reputation of airlines.”(Page1).
=> “Key words: perceived quality; corporate image; customer’s trust; corporate reputation” (Page1 line 14).
2. The purposes expressed in lines 40-45 at the end of introduction are not reached. This is a weak point that must be seriously reword for clarity. First, the theoretical model is not validated because some hypotheses will be rejected. Second, is partially reached because service quality is a multidimensional construct and only some dimensions are confirmed in the empirical study. Third, the article does not analysed mediation effects. For this point the authors could consider following Mallinckrodt et al. (2006) for instance.
Response: We thank the reviewer for pointing out this. Related part has been revised to state the study purpose precisely.
=> “Accordingly, the main purpose of current research is to reveal the influence mechanism of airline service quality to corporate reputation by using SERVQUAL and RQ (Reputation Quotient) measures.”(Page 2 line 42-44)
3. The final section devoted to corporate reputation finishes presenting a scale for measuring it with six dimensions and one of them is surprisingly quality of service (Lines 111-123). This is an incongruence because the theoretical model relates the perceived quality of the service with corporate reputation that includes at the same time the quality of the service.
Response: We truly appreciate your valuable comment. But in my opinion, service quality is essential to the survival of airlines, and the main purpose of current study is to investigate how airline service quality impacts airline corporate reputation. Therefore, it is reasonable to put the service quality in the most front of the research model and consider service quality as an important antecedent variable in the current study. Meanwhile, reputation quotients can be slightly modified in different settings, as the airline industry is unique, it is rational to remove service quality in the RQ measures as it was already set as an antecedent variable.
In order to state it more logically, we added a prior study that modified RQ in the setting of university as follows:
=> “For example, Del-Castillo-Feito, Blanco-González, & González-Vázquez employed reputation quotient to examine the relationship between image and reputation in the context of universities. And reputation quotient was measured through performance, innovation, social responsibility, services, governance and workplace climate, which was modified based on the research object. ” (Page 4 line 128-132)
4. Hypothesis 3, that is divided into 4 sub-hypotheses (Lines 184-187), forgets service quality (the first dimension in Lines 111-123), forgets leadership (only remains vision), and finally the hypothesis 3 forgets emotional appeal.
Response: Thanks for your keen observation. For service quality, we responded this question in the former response. For leadership, we were careless when double checking variable’s name, items about leadership were also asked in the questionnaire, so we revised all the words related to this variable into “leadership and version” (Page 5 line 189, 191,…). For emotional appeal, we are sorry to say that we did not consider this dimension when designing questionnaire because the construct is considered not to reflect the context of airline industry, it will be a limitation of this research.
5. Please, consider service quality as a second order construct or change the rationale of the manuscript to explain and justify what already is tested with the model.
Response: We truly appreciate your valuable comment. But airline is a unique setting, service quality is significant to airline industry. SERVQUAL is a common measure for service quality and we want to reveal which dimensions in the measures affect other variables (i.e., trust and image) most, so we think it is rational to consider each dimension as a single construct by adopting first-order approach to test the relationship between service quality with other variables.
6. I do not recommend explaining in detail the three methods to test discriminant validity when only the third one is giving the authors good results. I will offer directly the third method to test discriminant validity. Even maintaining the three methods, it is not logical to offer table 3 containing information that is not supporting discriminant validity. Please, remove table 3 and instead offer a table with information related to method number three.
Response: Thanks for your suggestions. Actually, it is a step by step process to test discriminant validity, besides it is a standard description method to mention three methods when testing discriminant validity. Table 3 provides all related values in discriminant validity testing, thus it is necessary to retain table 3. For method three, it only related to two variables (i.e., WE and VL), it is not necessary to make a table for it, so we described the calculate result in the manuscript according to prior studies (Song, Lee, Norman, & Han, 2012).
Song, H. J., Lee, C. K., Norman, W. C., & Han, H. (2012). The role of responsible gambling strategy in forming behavioral intention: An application of a model of goal-directed behavior. Journal of Travel Research, 51(4), 512-523.
7. Figure 2 has not good quality, please improve it.
Response: Figure 2 has been redrawn in manuscript (Page 12 line 331):
Minor changes
Line 29 – Please rewrite for clarity. May be full stop after “patronage” should be enough.
Response: The sentence has been revised as follows:
=> “One purpose of providing better quality is to meet customer’s expectation and retain customer patronage. It was general accepted that good service quality can lead to customer’s satisfaction and loyalty in prior studies.”(Page 2 line 27-29)
Lines 34-35 – This strong assertion needs at least one reference.
Response: We thank the reviewer for pointing out this. Citation has been added as follows:
=> “Reputation was considered as the most valuable intangible asset for a company to maintain a sustainable competitive advantage [11,12].”(Page 2 line 33-34)
Lines 38-40 – In my opinion, citation should be at the end of the sentence as follows: Herbing et al. stated that corporate reputation was built through its credible actions [14].
Response: Thanks for your keen observation. It has been revised as follows:
=> “Herbig, Milewicz and Golden stated that corporate reputation was built through its credible actions [14].”(Page 2 line 37-38)
Line 50 – Rewrite for clarity. May be including “the” in the sentence would be good: In which degree “the” provided service…
Response: Thanks for your keen observation. The sentence has been revised as follows:
=> “Perceived service quality refers to the discrepancy between customer’s subjective expectation and his/her actual perception of service performance, which was used to measure in which degree does provided service matches customer’s expectation [5,17].”(Page 2 line 47-49)
Line 56 – SERVQUAL is already a known scale, it was and it is. I think that the correct would be to say “is”.
Response: Thanks for your keen observation. The word has been modified as follows:
=> “SERVQUAL is a common scale to measure perceived service quality, which consists of five factors, namely tangibles, responsiveness, reliability, empathy and assurance. SERVQUAL was firstly put forwarded by Parasuraman, Zeithaml and Berry.”(Page 2 line 55)
Line 60-61 – Please explain the results obtained in citation number [22] because it seems to be relevant for this study.
Response: We thank the reviewer for pointing out this. Result of prior studies has been added as follows:
=> “Alkhatib and Migdadi identified the key operational determinants of airline service quality across regions, flight ranges, and destinations through multi-regression multi-level analysis. Results showed that the strongest impact of the operational determinants on airline service quality was in the Arab states region, while the weakest impact was in the Europe region, and there is no significant impact in the Africa region[22].”(Page 2 line 60-65)
Line 218-219 -Please explain the pilot test with details.
Response: Thanks for your keen observation. Pilot test was described as follows:
=> “A pilot test was conduct to airline managers, flight attendants, and graduate students during January, 2018. In pilot test, 50 valid questionnaires were collected, which helps to refine measurement items for clarity.”(Page 7 line 227-230)
Line 231 – Please explain the representativeness of the sample.
Response: Thanks for your keen observation. The explanation has been added as follows:
=> “Asiana Airlines was selected as the research object as it is one of two representative airline corporates in Korea (i.e., Korean Airlines and Asiana Airlines), with high popularity and good quality service.”(Page 7 line 237-239)
Line 234 – Please explain how do you detect insincere responses.
Response: Thanks for your keen observation. We checked whether respondents responded the questionnaire too casually and whether there was paradox respond when we coding the questionnaire.
Line 252 – I wonder whether this table is relevant in the paper considering that there are not analyses testing moderation effects of gender, education or income. In addition, in the table I wonder whether language could be corrected to avoid sexism. For instance, businessman could be also businesswoman and, may be, there is an alternative word for housewife.
Response: Thanks for your keen observation. Table 1 is to show the sample characteristics. Businessman is a general call for both businessman and businesswoman, so we think it is rational to use businessman here, because we just want to show the sample characteristics and do not analyze moderation effects or multi-group effects or T-test of gender and occupation. And housewife is a common occupation in Korean social and culture, we think it is reasonable to use this word to describe this occupation.
Thank you so much for your comments and the time you gave for the improvement of our manuscript.

Reviewer 2 Report
First of all, I would like to keep your attention to the use of the semicolons in the abstract. Please revise all the sentences and make sure that sentences are correct before resubmission.
SECTION 1: Introduction
I would like to keep the attention of the Authors about the classical structure that an introduction should have. As clarified in the "Instruction for Authors" of this journal: "The introduction should briefly place the study in a broad context and highlight why it is important. It should define the purpose of the work and its significance, including specific hypotheses being tested. The current state of the research field should be reviewed carefully and key publications cited. Please highlight controversial and diverging hypotheses when necessary. Finally, briefly mention the main aim of the work and highlight the main conclusions. Keep the introduction comprehensible to scientists working outside the topic of the paper."
Where is your added value? Why this work is important and why it should be published? Where is the structure of the paper? I think that probably these aspects should be stressed.
SECTION 3: Methodological approach
With regard to this section, I have some major issues. First of all, why do you decided to focus on Asiana Airline? In selecting a particular case it should be explained the reason.
I think that this section should be improved. You use a methodological approach but what is the reason? Is the most used, is the most recommended?
Moreover, about "the use of SERVQUAL in airline services" look for example at:
· Pakdil, F., & Aydın, Ö. (2007). Expectations and perceptions in airline services: An analysis using weighted SERVQUAL scores. Journal of Air Transport Management, 13(4), 229-237.
· Farooq, M. S., Salam, M., Fayolle, A., Jaafar, N., & Ayupp, K. (2018). Impact of service quality on customer satisfaction in Malaysia airlines: A PLS-SEM approach. Journal of Air Transport Management, 67, 169-180.
· Jiang, H., Baxter, G. S., & Wild, G. (2017). A study of China’s major domestic airlines’ service quality at Shanghai’s Hongqiao and Pudong International Airports. Aviation, 21(4), 143-154.
Another issue relates to the response rate. Are you sure that the presence of "a small gift" do not represent a behavioral distortion? It seems more like a commercial procedure and not an academic procedure. Could you provide me some academic papers or evidence about the use of this practice? Please consider this as a question to be addressed.
Finally, you should improve the section "references" (not up to date).
Author Response
We sincerely appreciate the reviewers’ comments and suggestions on the previous version of this manuscript. We have thoroughly studied them and have revised the manuscript accordingly. This report summarizes our responses to all the comments (in green for your convenience).
1. First of all, I would like to keep your attention to the use of the semicolons in the abstract. Please revise all the sentences and make sure that sentences are correct before resubmission.
Response: Thanks for your keen observation. Related part has been revised as follows:
=> “With the continuous increasing demand for airlines, competition among airlines has become serious as well. The purpose of this study is to empirically examine the causal relationships among perceived service quality, corporate image, customer’s trust, and corporate reputation of Asiana Airline in South Korea using SERVQUAL measures. An onsite survey was conducted for Korean passengers. The results of this study reveal that responsiveness and reliability significantly affect corporate image and customer’s trust, whereas tangibles, empathy and assurance are not significant antecedents of corporate image and customer’s trust. It is also found that corporate image and customer’s trust significantly affect corporate reputation. Based on these findings, theoretical importance and practical implications are discussed in this paper.”(Page1 line 3-12)
2. Where is your added value? Why this work is important and why it should be published? Where is the structure of the paper? I think that probably these aspects should be stressed.
Response: We thank the reviewer for pointing out this. Related part has been added in introduction as follows:
=> “Furthermore, it has been stated that corporate reputation was affected by corporate image and customer’s trust [15,16], while little research empirically integrated service quality and three important corporation aspects: corporate image, customers’ trust, and corporate reputation together to explore the process of corporate reputation’s building. Accordingly, the main purpose of current research is to reveal the influence mechanism of airline service quality to corporate reputation by using SERVQUAL and RQ (Reputation Quotient) measures.”(Page1 line 38-44)
3. Why do you decided to focus on Asiana Airline? In selecting a particular case it should be explained the reason.
Response: Thanks for your keen observation. The explanation has been added as follows:
=> “Asiana Airlines was selected as the research object as it is one of two representative airline corporates in Korea (i.e., Korean Airlines and Asiana Airlines), with high popularity and good quality service.”(Page 7 line 237-239)
4. You use a methodological approach but what is the reason? Is the most used, is the most recommended?
Response: Thanks for your question. SEM is the most used methodology to explore the relationship among various variables. The main purpose of the current study is to reveal the building mechanism of corporate reputation, which need to analyse variables that can affect corporate reputation. Thus, we adopt SEM in this study to examine the relationships among service quality, corporate image, customer’s trust and corporate reputation.
5. Another issue relates to the response rate. Are you sure that the presence of "a small gift" do not represent a behavioural distortion? It seems more like a commercial procedure and not an academic procedure. Could you provide me some academic papers or evidence about the use of this practice? Please consider this as a question to be addressed.
Response: Thanks for your keen observation. After returning the questionnaire, providing the respondent with a small gift is an acceptable way to express gratitude. It has been mentioned in some previous studies (Jim & Chen, 2006; Jörngården, Mattsson, & Von Essen, 2007).
Jörngården, A., Mattsson, E., & Von Essen, L. (2007). Health-related quality of life, anxiety and depression among adolescents and young adults with cancer: a prospective longitudinal study. European Journal of Cancer, 43(13), 1952-1958.
Jim, C. Y., & Chen, W. Y. (2006). Recreation–amenity use and contingent valuation of urban greenspaces in Guangzhou, China. Landscape and urban planning, 75(1-2), 81-96.
6. Please update the references.
Response: Thanks for your suggestion. References have been double check as required.
Thank you so much for your comments and the time you gave for the improvement of our manuscript.

Reviewer 3 Report
Reviewed study is interesting, used well known methodology and focus on case study however it is not strongly related to the sustainability. I propose the Authors to relate more clearly the study to the sustainability issue.
Introduction motivates the study well and provide necessary background.
The literature review is impressive and provide logical information flow to construct the hypothesis
Authors should provide time period of data collection.
Study results presentation is an excellent part of the paper.
I propose to separate the study discussion and conclusion. Add study limitations and future research trends.
Author Response
We sincerely appreciate the reviewers’ comments and suggestions on the previous version of this manuscript. We have thoroughly studied them and have revised the manuscript accordingly. This report summarizes our responses to all the comments (in purple for your convenience).
1. Authors should provide time period of data collection.
Response: We thank the reviewer for pointing out this. Related part has been mentioned as follows:
=> “With regard to a specific data collection process, an onsite intercept survey of Korean passengers who experienced flying on Asiana Airlines was conducted from March to May, 2018 in three international airports' lounges (Incheon International Airport, Gimpo International Airport, and Gimhae International Airport).”(Page7 line 234-237)
2. I propose to separate the study discussion and conclusion. Add study limitations and future research trends.
Response: Thanks for your suggestion. The last part has been revised as follows:
=> “5.1 Conclusion
The results of the current study indicate that, among the five service quality dimensions, responsiveness and reliability dimensions have a significant effect on corporate image and customer’s trust; whereas tangibles, empathy and assurance have an insignificant effect on corporate image and customer’s trust. In addition, both corporate image and customer’s trust positively influence four dimensions of corporate reputation. Thus, corporate image and customer’s trust appear to mediate some airline service quality and airline corporate reputation. Findings of this study confirms corporate image and customer’s trust as mediating factors between service quality and corporate reputation in airline industry, and findings can also be valuable in subsequent studies for further comprehending the building of airline corporate reputation.”(Page 12 line 350-359)
=> “5.3 Limitations and future research trends
Second, only four dimensions were used in the Reputation Quotient, emotional appears was not considered when measuring corporate reputation. Thus, a comparison between FSC (Full Service Carriers) and LCCs (Low Cost Carriers) can be conducted in future studies to get a comprehensive understanding of airline corporation reputation building process.”(Page14 line 431-435)
Thank you so much for your comments and the time you gave for the improvement of our manuscript.

Round 2
Reviewer 1 Report
The paper has improved after the first revision. However, new information has been included that must be reconsidered. I am refering the inclusion of "mediation" in conclusions (i.e. corporate image and customer trust appear to mediate...; Findings of this study confirms corporate image and customer trust as mediating factors between ....). As far as I know the authors have not calculated mediation effects and consequently they can not say "appear to mediate" because it is not scientific, and they can not say that the study confirms mediation because it has not being considered. My recommendation is rewording conclusions to avoid any reference to mediation effects or, much better for the final version of your work, presenting mediation analysis to demonstrate that mediation exists. An example for calculating mediation effects is here:
https://www.emeraldinsight.com/…/full/10…/SAMPJ-07-2018-0171
All the best
Author Response
Response to Reviewer 1 Comments
We sincerely appreciate the reviewer’s comment and suggestion on the previous version of this manuscript. We have thoroughly studied and revised the manuscript accordingly. This report summarizes our responses to the comments (in red for your convenience).
1. My recommendation is rewording conclusions to avoid any reference to mediation effects or, much better for the final version of your work, presenting mediation analysis to demonstrate that mediation exists.
Response: Thanks for your suggestions. All the parts related to mediation was deleted and revised as follows.
=> “Thus, findings of this study confirms that some of airline service quality affect corporate reputation through corporate image and customer trust.”(Page 12 line 360-362)
=> “Moreover, it was found that partial perceived service quality affects corporate reputation through corporate image and customer trust. This finding implies that customer’s perception of service quality directly contributes to generating or improving the airline corporate image and customer trust, and indirectly contributes to building airline corporate reputation. Thus, in practice, it is crucial to establish favorable corporate image of friendly, thoughtful, clean and reliable, as well as to gain customer trust by keeping promises and meeting expectations of customers. Because good service quality can establish favorable corporate reputation through corporate image and customer trust.”(Page 13 line 391-398)
Thank you so much for your comments and the time you gave for the improvement of our manuscript.

Round 3
Reviewer 1 Report
Thanks for taking into consideration my comments. In my opinion the article should be published in this new version, without taking into account the English language style because I don´t feel qualified to judge it.